# Polyanions provide selective control of APC/C interactions with the activator subunit

Arda Mizrak[1] & David O. Morgan [1]*

Transient interactions between the anaphase-promoting complex/cyclosome (APC/C) and its activator subunit Cdc20 or Cdh1 generate oscillations in ubiquitylation activity necessary to maintain the order of cell cycle events. Activator binds the APC/C with high affinity and exhibits negligible dissociation kinetics in vitro, and it is not clear how the rapid turnover of APC/C-activator complexes is achieved in vivo. Here, we describe a mechanism that controls APC/C-activator interactions based on the availability of substrates. We find that APC/C-activator dissociation is stimulated by abundant cellular polyanions such as nucleic acids and polyphosphate. Polyanions also interfere with substrate ubiquitylation. However, engagement with high-affinity substrate blocks the inhibitory effects of polyanions on activator binding and APC/C activity. We propose that this mechanism amplifies the effects of substrate affinity on APC/C function, stimulating processive ubiquitylation of high-affinity substrates and suppressing ubiquitylation of low-affinity substrates.

[1] Department of Physiology, University of California, San Francisco, CA 94143, USA. *email: David.Morgan@ucsf.edu

Regulatory systems in biology often depend on the transient formation of specific protein complexes. Dynamic interactions between protein partners must be achieved in the crowded and complex environment of the cell, where countless nonspecific collisions with other macromolecules can influence the rate of complex formation and disassembly. Numerous mechanisms are employed to enhance the specificity of protein interactions under these conditions, but our understanding of this problem is limited in part because studies of these systems often depend on analysis with purified components.

Dynamic and specific protein–protein interactions lie at the heart of the regulatory system that governs progression through the cell division cycle. One of the key components of this system is a ubiquitin ligase, the anaphase-promoting complex/cyclosome (APC/C), which triggers chromosome segregation and the completion of mitosis[1,2]. The APC/C is activated at specific cell-cycle stages by transient interactions with an activator subunit. Despite their central role in ordering mitotic events, the mechanisms that govern APC/C-activator interactions remain poorly understood.

The APC/C is a ubiquitin ligase (E3) of the RING family, which catalyzes the assembly of polyubiquitin chains on lysine side chains within specific substrates, leading to their degradation by the proteasome. The APC/C is composed of 13 subunits that form a stable 1.3 MDa core enzyme[3,4]. Its activation requires association with an activator subunit, either Cdc20 or Cdh1, which acts as a substrate receptor and also stimulates enzymatic activity. The two activators act in sequence to promote degradation of a series of substrates[5–13]. From prometaphase to anaphase of mitosis, the APC/C associates with Cdc20 to ubiquitylate securin and mitotic cyclins, driving sister-chromatid separation and the completion of anaphase. Cdc20 dissociates from the APC/C in late anaphase and is replaced by the second activator, Cdh1. Cdh1 maintains APC/C activity through late mitosis and during the subsequent G1, resulting in the sequential degradation of numerous Cdh1-specific substrates. When the cell enters the next cell cycle in late G1, Cdh1 dissociation inactivates the APC/C, thereby allowing cyclins and other APC/C targets to accumulate. Thus, transient association of the APC/C with two different activator subunits organizes the timing of cell-cycle events.

Cdc20 and Cdh1 are structurally related proteins containing a seven-blade β-propeller WD40 domain that is the primary site of substrate binding. This domain is anchored to the APC/C by sequences in flanking N-terminal and C-terminal regions. The N-terminal segment includes highly conserved interaction motifs, such as the C-box, that are critical for association with the APC/C core[13,14]. Upon binding the APC/C, the unstructured regions around the C-box form a helical bundle that interacts extensively with a cavity in the APC/C[15,16]. Binding of activator to the APC/C also depends on a dipeptide IR motif at the activator C-terminus, which interacts with a binding groove on the Cdc27/Apc3 subunit of the APC/C[17–21]. These N-terminal and C-terminal motifs are indispensable for activator binding, and mutation of either region abolishes APC/C-activator interactions.

APC/C substrates interact with the activator WD40 domain via short linear sequence motifs called degrons, of which the D box, KEN box, and ABBA motif are the most common[18,22–29]. Cooperative binding of multiple degron sequences to the activator subunit provides the affinity required for efficient and processive ubiquitylation, and the affinity of degron binding is likely to influence the timing of degradation in vivo[26,30,31]. Each of the three major degrons binds to a specific binding site on the activator WD40 domain. The KEN box and ABBA motif interact directly with the WD40 domain, while the D box interacts with both the WD40 domain and the adjacent core APC/C subunit Apc10/Doc1 (refs [23,25,27,32–36]). Bivalent D box binding enhances APC/C–activator interactions by bridging the WD40 domain to the

APC/C, providing additional interaction surfaces that stabilize the position of the activator on the APC/C[3,15,20,37–39].

To gain more insight into the molecular mechanisms of APC/C–activator interactions, we analyze the dissociation dynamics of activators in vitro. Using biochemical methods, we show that nucleic acids and other polyanions in cell lysates are able to rapidly dissociate activators from the APC/C. Interestingly, substrate D box interactions between activator and the core APC/C reduce activator dissociation by polyanions, providing a mechanism that enhances activator binding when the enzyme is occupied with high-affinity substrate.

## Results

**Cell lysates dissociate Cdh1 and Cdc20 from the APC/C.** The transient binding of activators to the APC/C during the cell cycle indicates that the activator dissociation rate in the cell must be sufficiently high to allow loss of activator from the APC/C over short time scales of minutes (Fig. 1a). To investigate the dynamics of activator binding, we developed an assay to measure activator dissociation from the budding yeast APC/C in vitro (Fig. 1b). We used translation in vitro to prepare $^{35}$S-labeled activator proteins, which were incubated with APC/C that had been immunopurified on magnetic beads from asynchronous cells carrying TAP-tagged APC/C subunit Cdc16. Following extensive washing and dilution, the amount of bound activator was then measured over time to estimate an activator dissociation rate (Fig. 1b).

Cdh1 and Cdc20 both exhibited negligible dissociation in 30 min (Fig. 1c, left panels), suggesting that activators bind the APC/C with high affinity. These results are consistent with the extensive binding interfaces seen in structural studies of APC/C-activator complexes[3], but they are not consistent with the rapid dissociation that occurs in vivo. We hypothesized that active mechanisms might exist in the cell to promote activator dissociation. Indeed, we found that incubation of APC/C$^{Cdh1}$ or APC/C$^{Cdc20}$ with whole lysates of budding yeast cells promoted activator dissociation (Fig. 1c, right panels). This effect was not due to disassembly of the APC/C or proteolysis of the APC/C or activator (Supplementary Fig. 1a, b). Similar results were obtained with APC/C that was tagged at another APC/C subunit, Apc1, suggesting that activator dissociation is not due to the instability of the tagged Cdc16 subunit (Supplementary Fig. 1c). Dissociation rate depended on lysate concentration, and robust activator dissociation was achieved with cell lysates containing over 1 mg/ml of protein (Fig. 1d).

Cdc20 and Cdh1 activate the APC/C at different stages of the cell cycle. To test whether dissociation activity is temporally regulated in a similar fashion, we compared lysates from cells arrested with mating pheromone (G1 arrest) or microtubule-depolymerizing drug nocodazole (metaphase arrest). Dissociation was promoted equally by these two lysates (Supplementary Fig. 1d), suggesting that the dissociation activity does not change during the cell cycle.

ATP hydrolysis is the major energy source for protein remodelers, which are capable of pulling subunits of protein complexes apart. To investigate whether activator dissociation in cell lysates is ATP-dependent, we used gel filtration to remove molecules smaller than 5 kDa from cell lysates. Depleted lysates showed no significant dissociation activity, but activity was restored by addition of 5 mM ATP (Fig. 1e). ATP alone did not promote activator dissociation (Fig. 1c, left panels). Addition of AMP-PNP also restored activity to depleted lysates (Fig. 1e), indicating that ATP hydrolysis is not required for the activity.

Stimulation of the dissociation activity by AMP-PNP ruled out canonical chaperone systems, such as Hsp70, Hsp90, and Hsp104, which require ATP hydrolysis to carry out their functions. The lack of a requirement for ATP hydrolysis also ruled out Cdh1

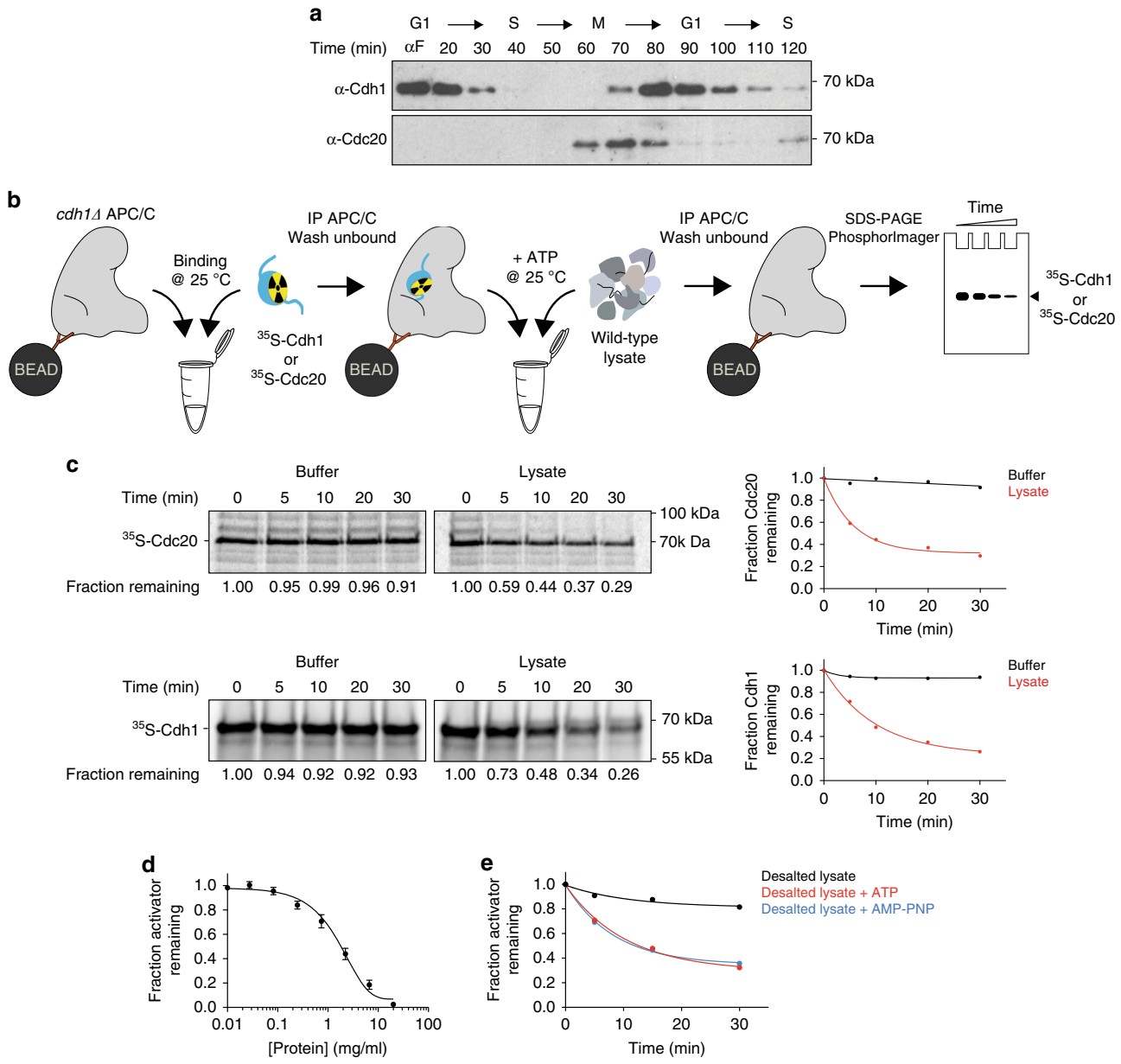

**Fig. 1 A biochemical activity in yeast cell lysates dissociates activators from the APC/C. a** Cells expressing TAP-tagged Cdc16 were arrested at G1 with 1 µg/ml α-factor for 3 h and released into YPD. APC/C was immunoprecipitated at the indicated time points and western blotted for Cdh1 (upper) and Cdc20 (lower). Uncropped western blots are provided in the Source Data file. **b** This schematic illustrates our APC/C-activator-binding assay. cdh1Δ APC/C (TAP-tagged on the Cdc16 subunit) was immunopurified on magnetic beads and incubated with $^{35}$S-Cdc20 or $^{35}$S-Cdh1 produced by translation in vitro. After removing unbound activators by washing, buffer or yeast lysate (2.5 mg/ml protein concentration) was added in the presence of 5 mM ATP. At various times, APC/C was pulled down from the reaction mix and the amount of bound activator determined by SDS–PAGE and PhosphorImager analysis. Due to the extreme dilution of APC/C-activator complexes on the beads, rebinding of dissociated activator is expected to be negligible under these conditions. **c** As described in panel **b**, we measured dissociation of radiolabeled Cdc20 (top) or Cdh1 (bottom) from immobilized APC/C over the indicated time course (at 25 °C). Buffer control contains lysis buffer and 5 mM ATP. Fraction remaining was measured by calculating the ratio of activator signal at indicated time points to zero time point signal. Results were plotted and fitted to an exponential one phase decay equation in GraphPad Prism. Similar results were observed in several independent experiments. Source data are provided in the Source Data file. **d** Cdh1 dissociation reactions were performed as in panel **c** in the presence of serially diluted yeast lysate, supplemented with 5 mM ATP. Remaining activator after 45 min is plotted as a function of protein concentration. Data indicate means (±SEM) from three independent experiments. Source data are provided in the Source Data file. **e** Yeast lysates were subjected to gel filtration in lysis buffer to remove molecules smaller than ~5 kDa. Desalted lysates were supplemented with buffer (black), 5 mM ATP (red), or 5 mM AMP-PNP (blue) prior to the Cdh1 dissociation reaction. These results are representative of multiple experiments. Source data are provided in the Source Data file.

phosphorylation, which is known to inhibit its binding to the APC/C[10,12]. We further narrowed the possible candidates by estimating the size of the activity by gel filtration chromatography. The activity eluted with a broad peak of 80–100 kDa (Supplementary Fig. 2a).

This molecular size eliminated the possibility that activator dissociation was due to large molecular machines, such as the proteasome or the chaperonin CCT, both of which are thought to influence activator binding to the APC/C[40–43].

**Characterization of the activator dissociation activity**. We attempted to isolate the dissociation activity from yeast lysates using a variety of chromatographic methods and other purification approaches. During the course of these studies, we discovered that the activity is resistant to boiling. Incubation of cell lysates at 95 °C for 10 min led to precipitation of most proteins in the lysate, but the dissociation activity remained in solution (Fig. 2a). We then incorporated heat treatment as a step in an effective purification scheme, as follows (Fig. 2b). We applied cell lysate to a hydroxyapatite column, from which the activity could be eluted with 100 mM phosphate. The eluate was boiled, and the supernatant was dialyzed and re-applied to the hydroxyapatite column. The activity no longer bound to the column but was instead collected in the flow-through fraction (Fig. 2c). This method resulted in a high yield of activity and a high degree of purity. The flow-through fraction of this preparation was used for further characterization of the activity.

This fraction contained abundant activity despite having a very low protein concentration, leading us to question whether the dissociation activity is composed of protein or other macromolecules. Indeed, activity remained after treatment with the general protease Proteinase K (Fig. 2d), which led to the degradation of all detectable protein in the preparation (Supplementary Fig. 2b). However, treatment with a general nuclease, Benzonase, diminished the activity (Fig. 2d). Purification and analysis of the nucleic acids in the preparation revealed that the major nucleic acid species were about 80 nucleotides in length, and these species disappeared after RNase A treatment (Fig. 2e). We then purified the predominant RNA species and found that they removed Cdh1 from the APC/C in the presence of ATP, and RNase A treatment abolished the activity (Fig. 2f). These results argued that the RNA component is necessary and sufficient to dissociate activator subunit from the APC/C in an ATP-dependent manner.

**Polyanions promote activator dissociation**. To further understand what types of RNA molecules provide the dissociation activity, we cloned and sequenced the major RNA species in our active preparation, revealing that the major RNAs in the preparation were tRNAs and rRNA fragments (see the "Methods" section). tRNA$^{Gln}$

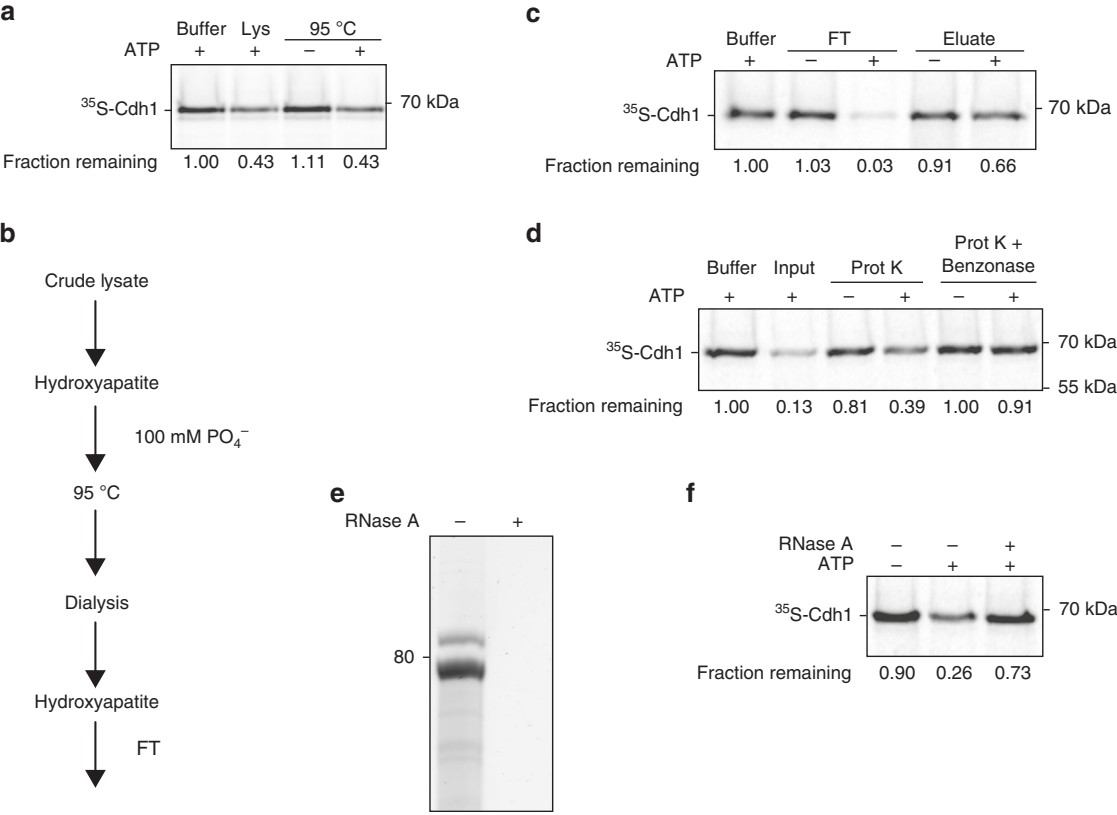

**Fig. 2 RNA in cell lysate is necessary and sufficient for activator dissociation. a** Cdh1 dissociation from APC/C was measured after 45 min in the presence of lysis buffer or yeast lysate, either untreated (Lys) or incubated at 95 °C for 10 min. Prior to the experiment, boiled lysate was gel filtered and concentrated five-fold. Dissociation reactions were performed with or without 5 mM ATP as indicated. Uncropped autoradiograph and source data are provided in the Source Data file. **b** Purification steps in the preparation of the dissociation activity used in panels **c–f**. Dissociation activity was found in the flow-through (FT) of the second hydroxyapatite column, which was also eluted with 100 mM phosphate. **c** Flow-through (FT) and eluate fractions of the last hydroxyapatite step were tested in a Cdh1-dissociation reaction. Fractions were buffer-exchanged into reaction buffer containing 2.5 mM MgCl₂. 5 mM ATP was added as indicated. The vast majority of the dissociation activity was found in the flow-through fraction. Uncropped autoradiograph and source data are provided in the Source Data file. **d** The hydroxyapatite flow-through fraction was incubated with buffer or Proteinase K (Prot K—6 U/ml) at 37 °C for 1 h. Proteinase K was heat inactivated at 85 °C for 10 min and further inhibited with 1 mM PMSF. After Proteinase K treatment, samples were incubated with buffer or Benzonase (2000 U/ml) at 37 °C for 1 h. Samples were supplemented with 5 mM ATP as indicated. Uncropped autoradiograph and source data are provided in the Source Data file. **e** Nucleic acid species in the flow-through fraction were extracted with phenol–chloroform, separated by a 10% TBE urea polyacrylamide gel and stained with SYBR Safe. RNase A treatment (0.2 mg/ml) was performed at 37 °C for 30 min (right lane).
**f** Phenol–chloroform-extracted RNA species from panel **e** were tested for Cdh1 dissociation with or without 3 mM ATP in the presence of a reaction buffer containing 2.5 mM MgCl₂. RNase A treatment (0.2 mg/ml) was performed at 37 °C for 30 min (right lane). Uncropped autoradiograph and source data are provided in the Source Data file.

and tRNA$^{Ser}$ were particularly abundant in the sample. To test their activity directly, tRNA$^{Gln}$ and tRNA$^{Ser}$ were transcribed in vitro and gel purified. Both species removed Cdh1 from the APC/C in the presence of ATP, with half-maximal dissociation occurring at concentrations of 0.7 and 0.9 μM, respectively (Fig. 3a).

To determine if the activity was specific for tRNA or could be seen with any RNA species, we transcribed a pool of random RNA sequences of the same length as the glutamine tRNA used above. These RNA molecules removed activator from the APC/C at similar concentrations as tRNA (0.9 μM, Fig. 3a), suggesting that activator dissociation could be achieved by a variety of RNA sequences. To determine if this mechanism is a general function of nucleic acids, we synthesized two random, complementary 75-base single-stranded DNA oligonucleotides. Both sequences dissociated the activator subunit from the APC/C even more efficiently than RNA, with half-maximal concentrations of 0.2 and 0.6 μM (Fig. 3b). These results suggest that nucleic acids, independent of their sequence, are capable of promoting dissociation of activators from the APC/C.

The above experiments were performed with single-stranded nucleic acid polymers. Compared to double-stranded species, single-stranded nucleic acids exhibit more flexibility and heterogeneity in secondary structure. Slight variations in the dissociation activity of the sequences we tested (particularly differences

between two complementary DNA oligonucleotides) suggest that nucleic acid structure and flexibility could play a role in disruption of APC/C-activator interactions. We explored this issue further by annealing the two complementary 75-nucleotide DNA oligonucleotides to reduce their flexibility and structural complexity. This double-stranded DNA dissociated activators less efficiently than either single strand (1 μM, Fig. 3c). Thus, efficient dissociation might be enhanced by the flexibility or secondary structures of single-stranded nucleic acids.

We also tested the effect of nucleic acid length on activator dissociation. A 75-base single-stranded DNA molecule was more than five-fold (half-maximal concentration 0.16 μM) more potent in activator dissociation than a 50-base oligonucleotide (1.8 μM). A 25-base oligonucleotide had little effect even at high concentrations (Fig. 3d). The higher activity of longer polymers could result from their greater flexibility and secondary structures. Activator dissociation might also depend on nucleic acid interactions with multiple distant sites on the APC/C and/or activator.

Given that both DNA and RNA can efficiently remove activators from the APC/C independent of nucleotide sequence, we next investigated if the negatively charged phosphate backbone of nucleic acids is responsible for activator dissociation. We tested the activity of polyphosphate, long chains of phosphate molecules that are found at high concentrations in cells from

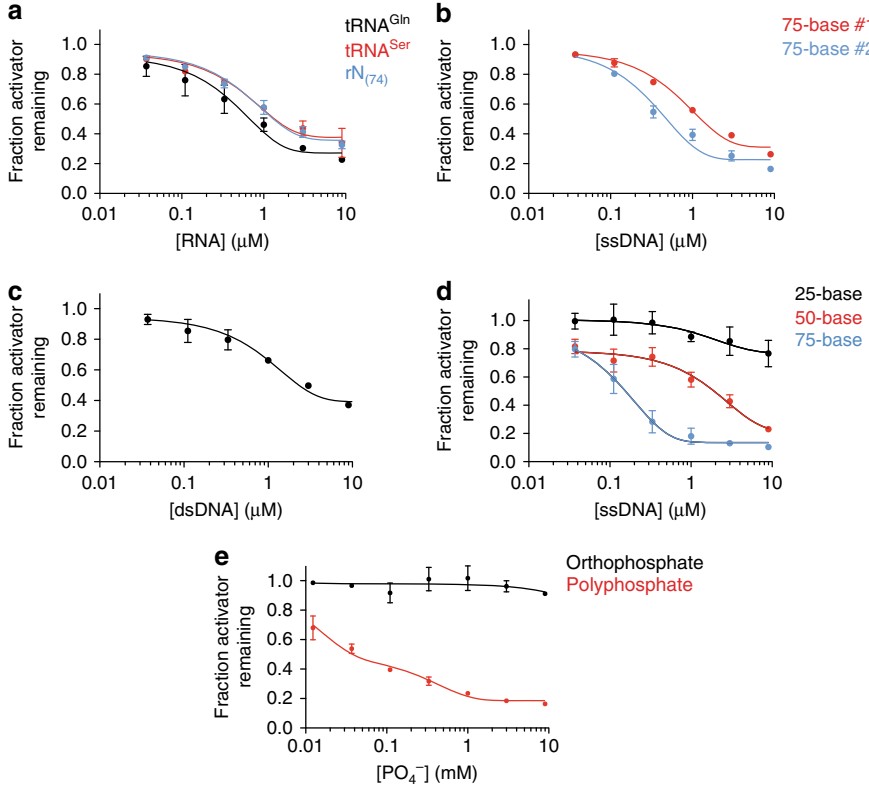

**Fig. 3 Nucleic acids and polyphosphate promote activator dissociation. a** RNA sequences for glutamine tRNA (tRNA$^{Gln}$), serine tRNA (tRNA$^{Ser}$), or a degenerate 74-nucleotide long RNA pool (rN$_{(74)}$) were transcribed in vitro, purified, and incubated in Cdh1 dissociation reactions for 45 min. Fraction of Cdh1 remaining at zero RNA concentration was not plotted on the log scale and taken as 1.0. Data indicate means (±SEM) from two independent experiments. Source data are provided in the Source Data file. **b, c** Cdh1 dissociation reactions were performed in the presence of: **b** two random 75-base ssDNA oligos complementary to each other; or **c** a double-stranded DNA formed by annealing the two single-stranded DNAs from panel **b**. See Supplementary Table 1 for sequences. Reactions were supplemented with 2.5 mM MgCl$_2$ and 3 mM ATP. Data indicate means (±SEM) from two independent experiments. Source data are provided in the Source Data file. **d** Single-stranded DNA oligonucleotides of varying lengths were tested for Cdh1 dissociation in the presence of 2.5 mM MgCl$_2$ and 3 mM ATP. The 75-base ssDNA used here and in all subsequent experiments is the 75-base ssDNA #2 in panel **b**. Data indicate means (±SEM) from three independent experiments. Source data are provided in the Source Data file. **e** Cdh1 dissociation reactions were performed at the indicated concentrations of orthophosphate (black) or polyphosphate (red) in a reaction buffer containing 3 mM ATP and no magnesium. Data indicate means (±SEM) from three independent experiments. Source data are provided in the Source Data file.

every branch of life[44]. In addition to providing energy storage, polyphosphate is believed to alter protein structure and stability through extensive electrostatic interactions with protein surfaces[45]. We found that a mixed length polyphosphate preparation (45–160 phosphates per chain) was capable of dissociating activators, while similar concentrations of orthophosphate had no effect (Fig. 3e). Polyphosphate concentrations required for activator dissociation were higher than those observed with nucleic acids, even if nucleic acid concentration is expressed in terms of total phosphate concentration. We suspect that this reduced potency is due to heterogeneity in polyphosphate chain length. Taken together, our results show that biological polymers containing long chains of negatively charged phosphates are potent catalysts of APC/C-activator dissociation.

**ATP promotes activator dissociation by sequestering $Mg^{2+}$.** Nucleic acids require ATP at millimolar concentrations for efficient activator dissociation (Supplementary Fig. 3a). ATP alone, even at high concentrations, is not sufficient to dissociate activators (Supplementary Fig. 3a), and its hydrolysis is not necessary (Fig. 1e). To explore the role of ATP further, we tested whether other nucleoside phosphates could substitute for ATP. The purified RNA preparation from the flow-through fraction of Fig. 2b displayed dissociation activity when supplemented with 3 mM ATP, AMP-PNP or GTP, and less effectively with ADP, but not with AMP (Fig. 4a). We hypothesized that dissociation might simply require molecules containing two adjacent phosphates. Consistent with this possibility, we found that inorganic pyrophosphate and tri-phosphate promoted the ability of DNA to dissociate the activator (Fig. 4b).

Phosphates on nucleic acids, polyphosphate chains, and free ATP interact with divalent cations such as $Mg^{2+}$. If polymeric negative charges are critical for the ability of nucleic acids and polyphosphate to dissociate activators, then magnesium ions could neutralize this charge and thereby block activator dissociation. Our dissociation reactions contain 2.5 mM $Mg^{2+}$ ions, and we therefore hypothesized that millimolar concentrations of ATP promote activator dissociation by sequestering these ions. Consistent with this hypothesis, 2.5 mM $Mg^{2+}$ inhibited DNA-dependent or polyphosphate-dependent activator dissociation in the absence of ATP (Fig. 4c–f; Supplementary Fig. 3b). In the absence of $Mg^{2+}$, addition of ATP did not increase polyphosphate-dependent Cdh1 dissociation and only slightly enhanced DNA-dependent dissociation (Fig. 4d, f). In reactions containing both ATP and $Mg^{2+}$, we observed that the inhibitory effect of $Mg^{2+}$ occurred only when it was in molar excess of ATP (Fig. 4g, Supplementary Fig. 3c). Interestingly, equimolar ATP did not completely abolish the inhibitory effects of $Mg^{2+}$ on polyphosphate-dependent Cdh1 dissociation (Fig. 4e, f). One plausible explanation is that polyphosphate, due to the close spacing of its phosphates, has a high affinity for $Mg^{2+}$ that is similar to that of ATP, and ATP is therefore a less effective competitor for $Mg^{2+}$ binding. In sum, our findings suggest that phosphate residues on ATP sequester $Mg^{2+}$ ions and thereby enhance the negative charge of phosphate-containing polymers, which is required for activator dissociation.

**Polyanions enhance APC/C substrate selectivity.** The APC/C ubiquitylates its targets only when bound by an activator, so dissociation of activators by polyanions should reduce APC/C activity. We tested this possibility by measuring APC/C ubiquitylation activity toward radiolabeled Pds1/securin in vitro. As predicted, incubation of APC/C$^{Cdh1}$ with single-stranded DNA prior to the ubiquitylation reaction reduced Pds1 ubiquitylation (Fig. 5a).

To better understand APC/C activity in the presence of DNA, we simultaneously measured Pds1 ubiquitylation and activator dissociation at various DNA concentrations. Even after just 2.5 min of incubation with DNA, Pds1 ubiquitylation by APC/C$^{Cdh1}$ was almost completely inhibited at DNA concentrations of 1 μM or higher (Fig. 5b, c), despite the fact that a significant amount of Cdh1 remained bound to the APC/C (Fig. 5b, d). After 30 min, ubiquitylation activity remained inhibited and was accompanied by more Cdh1 dissociation (Fig. 5c, d). These results revealed that inhibition of APC/C activity by polyanions is not due simply to activator dissociation. Instead, they suggest that inhibition occurs in two steps: rapid inhibition of substrate ubiquitylation followed by a slower process of activator dissociation.

We next analyzed the effect of DNA on the ubiquitylation of a selection of substrates, using Cdh1 or Cdc20 as activator. We tested three substrates: Pds1 and S-phase cyclin Clb5, both well-established substrates of APC/C$^{Cdc20}$ in metaphase; and Hsl1, a protein that is ubiquitylated primarily by APC/C$^{Cdh1}$ in late mitosis but is also an excellent substrate for APC/C$^{Cdc20}$ in vitro[15,37]. As before, we observed that APC/C$^{Cdh1}$ activity toward Pds1 was potently inhibited by single-stranded DNA, with a half-maximal DNA concentration of 0.2 μM (Fig. 5e). Interestingly, Pds1 ubiquitylation by APC/C$^{Cdc20}$ was relatively resistant to inhibition (Fig. 5e). Results with the S-phase cyclin Clb5 displayed a similar trend, with greater inhibition of activity with APC/C$^{Cdh1}$ and resistance to DNA with APC/C$^{Cdc20}$ (Fig. 5f). Finally, DNA only slightly reduced ubiquitylation of Hsl1 by APC/C$^{Cdc20}$ and had no effect on Cdh1-dependent activity (Fig. 5g).

The three substrates we tested are likely to have different affinities for different APC/C-activator complexes. Substrate affinity enhances processivity of the ubiquitylation reaction: a slower dissociation rate (and thus higher affinity) generally results in a greater number of ubiquitins attached during a single binding event[31–33]. Pds1 and Clb5 are both more processively modified by APC/C$^{Cdc20}$ than APC/C$^{Cdh1}$ (ref. [26]) (Fig. 5e, f), suggesting that they have greater affinity for Cdc20 than Cdh1. Hsl1 is an unusual substrate that is modified with high processivity with either activator (Fig. 5g), and it is known to contain an exceptionally high-affinity D box that has been used extensively in structural studies of the APC/C[3,15,46,47]. Together, these results suggest that DNA is a less effective inhibitor of reactions with substrates that bind with high affinity.

**D box binding inhibits polyanion-mediated dissociation.** Our evidence that high-affinity substrate ubiquitylation is resistant to inhibition by DNA led us to test the possibility that high-affinity substrate binding also blocks the effect of DNA on activator dissociation. We purified a fragment of Hsl1 (aa 667–872) that contains the D box and the KEN box and tested the effect of Hsl1 binding on single-stranded DNA-mediated dissociation. Hsl1 blocked DNA-driven dissociation of Cdh1 from the APC/C (Fig. 6a).

The D box sequence is known to bridge the WD40 domain of the activator subunit to the core APC/C subunit Apc10, thereby enhancing activator binding to the APC/C[20,32–37]. Consistent with this idea, we found that a peptide containing only the Hsl1 D box was able to block the effects of DNA and polyphosphate on activator dissociation (Fig. 6b; Supplementary Fig. 4a), while mutant D box or KEN box peptides had no effect (Fig. 6b).

Our previous results suggested that polyanionic inhibition of APC/C$^{Cdh1}$ activity toward Pds1 or Clb5 (Fig. 5e, f) results because their D boxes exhibit lower affinity for Cdh1 and their binding to the APC/C is therefore more easily disrupted by polyanions. We explored this possibility further by comparing the ability of Hsl1 and Pds1 D boxes, at varying concentrations, to

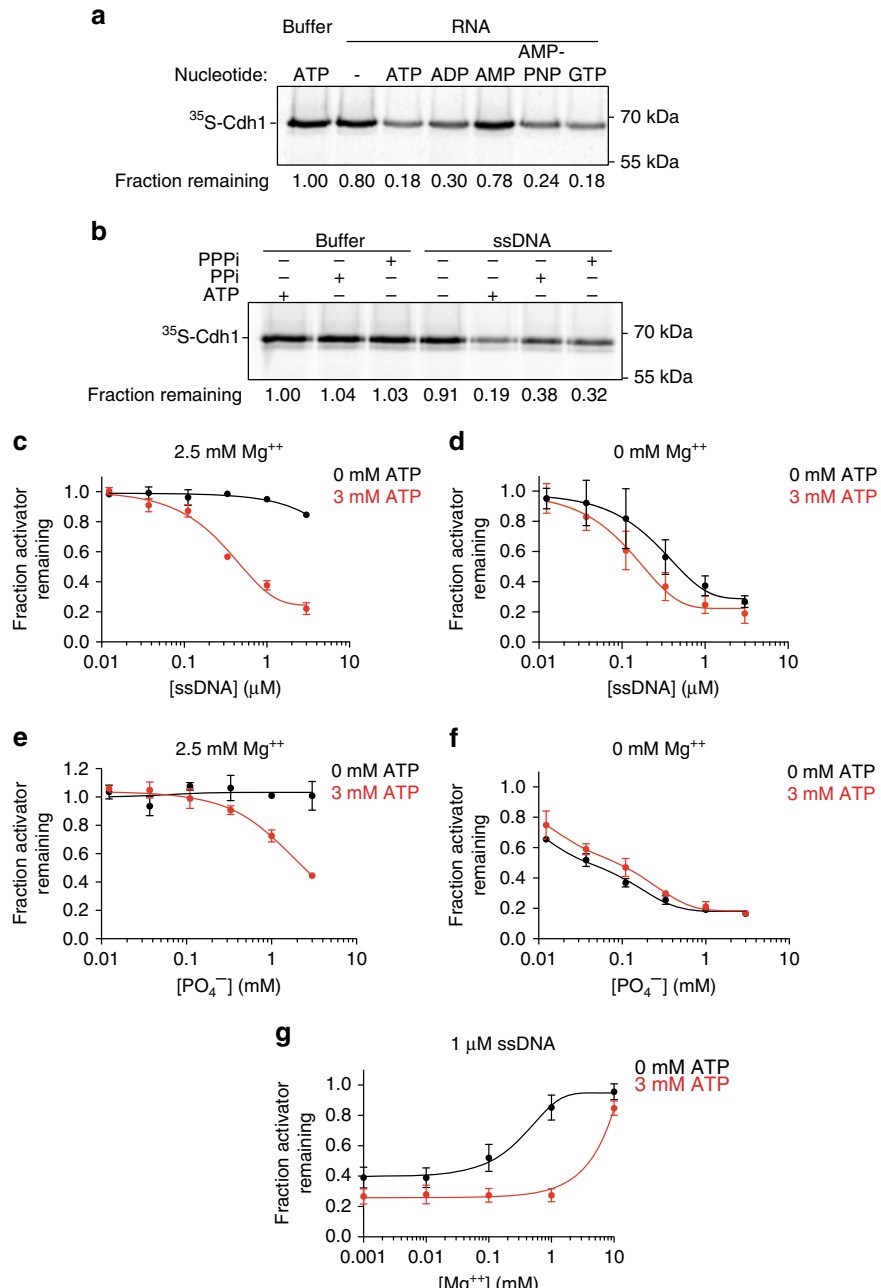

**Fig. 4 ATP promotes activator dissociation by sequestering magnesium ions. a** The effect of the indicated nucleotides on Cdh1 dissociation was tested using phenol–chloroform extracted RNA species from the hydroxyapatite flow-through fraction (Fig. 2b). Reactions were supplemented with 2.5 mM $MgCl_2$ and 3 mM of the indicated nucleotides. Uncropped autoradiograph and source data are provided in the Source Data file. **b** 3 mM ATP, inorganic pyrophosphate (PPi), or tri-phosphate (PPPi) were tested in a Cdh1-dissociation assay with reaction buffer or 1 µM 75mer ssDNA oligonucleotide. Reactions contain 2.5 mM $MgCl_2$. Uncropped autoradiograph and source data are provided in the Source Data file. **c–f** A 75mer ssDNA oligonucleotide **c, d** or polyphosphate **e, f** was tested in the Cdh1-dissociation assay in the presence **c, e** or absence **d, f** of 2.5 mM $MgCl_2$. Reactions were supplemented with 0 mM (black) or 3 mM ATP (red). Data indicate means (±SEM) from two independent experiments. Uncropped autoradiographs and source data are provided in the Source Data file. **g** The inhibitory effect of various $MgCl_2$ concentrations was analyzed in Cdh1-dissociation reactions. Reactions were performed using 1 µM 75mer ssDNA with (red) or without (black) 3 mM ATP. Data indicate means (±SEM) from two independent experiments. Uncropped autoradiographs and source data are provided in the Source Data file.

block activator dissociation by DNA. The Hsl1 D box peptide inhibited most activator dissociation at 0.4–4 µM, whereas equivalent inhibition of activator dissociation required 10–50-fold higher concentrations of the Pds1 D box peptide (Supplementary Fig. 4b, c). Similar differences were observed when we tested the effects of different D box concentrations on activator dissociation over a range of DNA concentrations (Fig. 6c, d). These results support the possibility that Hsl1 and Pds1 have

different affinities for APC/C[Cdh1], resulting in differing resistance to the effects of DNA. Also consistent with this possibility, we found that a Pds1 fragment (1-110aa) containing the D box effectively inhibited the dissociation of Cdc20 (Supplementary Fig. 4d). Together with our studies of ubiquitylation reactions in Fig. 5, these results argue that a high-affinity D box interaction holds the activator in a position that blocks the effects of polyanions on substrate ubiquitylation and activator dissociation.

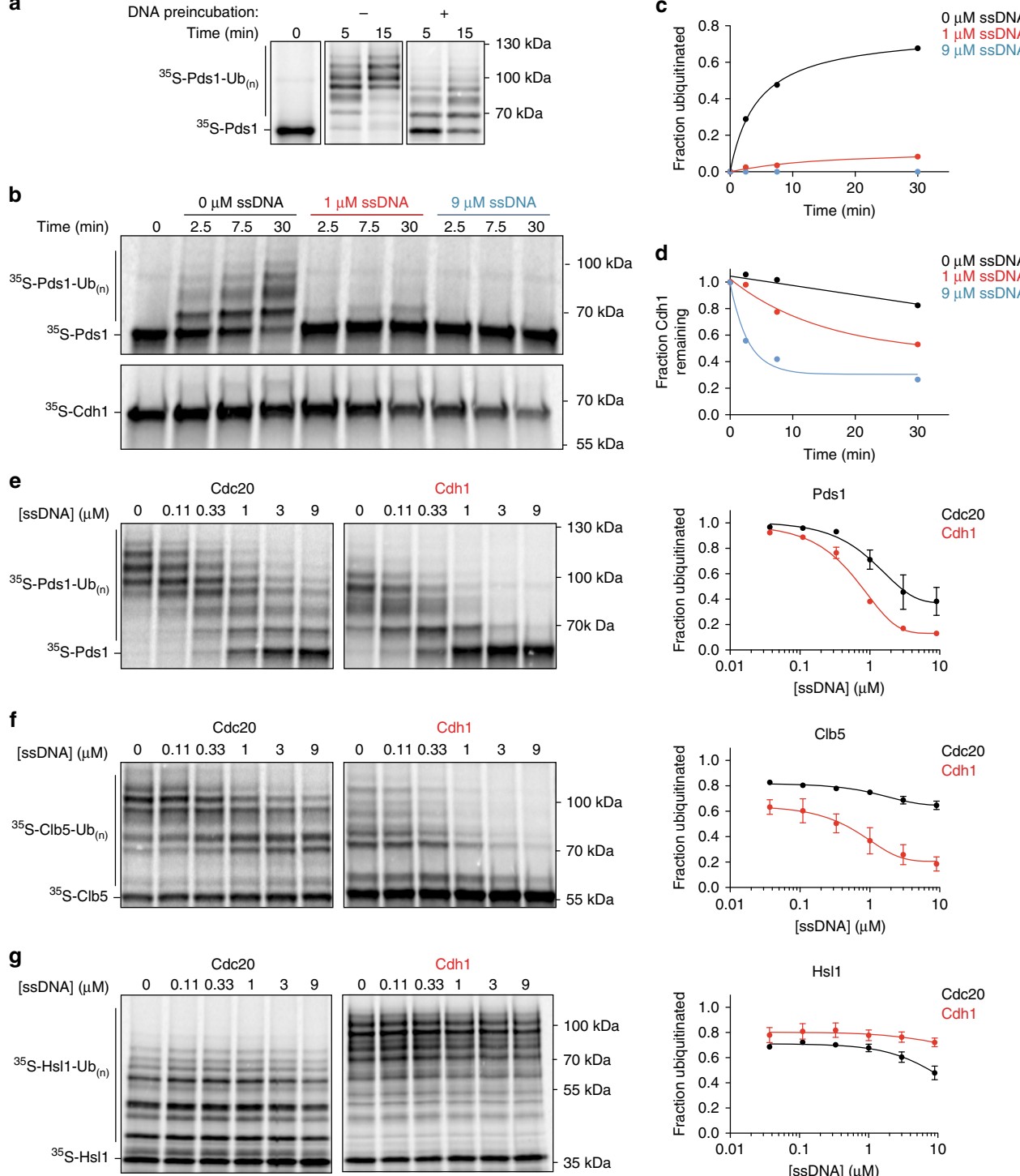

## Discussion

Precise temporal control of APC/C activity during the cell cycle relies on short-lived interactions between the APC/C and the activator subunit. Our studies suggest that the transient binding of activator to the APC/C is unlikely to result from spontaneous dissociation, which is negligible in vitro. Instead, rapid activator dissociation depends on abundant cellular polyanions, such as DNA, RNA, and polyphosphate, which also inhibit APC/C activity. The inhibitory effects of polyanions are blocked by strong substrate binding, providing a mechanism to selectively enhance ubiquitylation of high-affinity substrates.

There is growing evidence that negatively charged polymeric biomolecules influence the structural dynamics of proteins in the cell. Nucleic acids and polyphosphate exhibit chaperone-like properties, accelerate protein folding, and serve as anti-aggregation agents by solubilizing a variety of protein aggregates in vitro[48–50]. Long stretches of phosphate residues also inhibit polymerization of multimeric proteins and stabilize soluble structures[51,52]. Our work reveals that polyanions also influence the dynamics of APC/C-activator interactions, specifically disrupting APC/C-activator binding, while leaving the multi-subunit APC/C core intact.

**Fig. 5 Polyanions modulate APC/C activity toward different substrates. a** APC/C was immunopurified on beads and incubated with unlabeled Cdh1 (translated in vitro). Following washing, APC/C$^{Cdh1}$ was incubated 30 min at 25 °C with buffer (−) or 1 µM 75mer ssDNA (+) in the presence of 3 mM ATP. After two wash steps, $^{35}$S-Pds1 substrate was added to the APC/C$^{Cdh1}$ and ubiquitylation reactions were performed at 25 °C with methylated ubiquitin and Ubc4 as the E2. Reactions were terminated at the indicated times and analyzed by SDS–PAGE and PhosphorImager. Uncropped autoradiograph is provided in the Source Data file. **b** APC/C was purified from cells carrying Apc1-GFP using anti-GFP on magnetic beads. Ubiquitylation reactions were performed in two parallel experiments using $^{35}$S-Pds1 and unlabeled Cdh1 to monitor Pds1 ubiquitylation (upper panel) or $^{35}$S-Cdh1 and unlabeled Pds1 to detect activator dissociation (lower panel). 75mer ssDNA was added to the reactions at the indicated concentrations, with 3 mM ATP. Ubiquitylation reactions were terminated at the indicated times and analyzed by SDS–PAGE and PhosphorImager. For the $^{35}$S-Cdh1-dissociation reactions, immobilized APC/C was washed prior to analysis. Uncropped autoradiographs are provided in the Source Data file. **c**, **d** Quantification of results in panel **b**: **c** Fraction of ubiquitylated $^{35}$S-Pds1 was measured by calculating the ratio of ubiquitylated substrate to total substrate in the reaction. **d** $^{35}$S-Cdh1 dissociation was calculated by measuring $^{35}$S-Cdh1 that remained bound to the APC/C as in previous experiments. Source data are provided in the Source Data file. **e**–**g** Immunopurified APC/C was pre-bound with unlabeled Cdc20 (left) or Cdh1 (right) and then used in ubiquitylation reactions with one of three substrates (**e** $^{35}$S-Pds1; **f** $^{35}$S-Clb5; **g** $^{35}$S-Hsl1(aa 667-872)). Reactions were performed in the absence or presence of the indicated concentrations of the 75mer ssDNA, plus 2.5 mM MgCl$_2$ and 3 mM ATP. Ubiquitylation reactions were performed at 25 °C for 10 min (Hsl1) or 40 min (Pds1 and Clb5). Quantification of substrate ubiquitylation is provided in the panels at right. Data indicate means (±SEM) from two independent experiments. Uncropped autoradiographs and source data are provided in the Source Data file.

We found that submicromolar concentrations of RNA and DNA promote APC/C-activator dissociation, while polyphosphate acts at submillimolar concentrations. These concentrations are far lower than RNA and polyphosphate concentrations inside the cell. tRNA alone can be found in cells at a concentration of ~200 µM, while polyphosphate concentration is estimated to be as high as 100 mM (refs [44,53]). Even if these molecules are partially neutralized by positively charged ions, such as Mg$^{2+}$ and other charged macromolecules in the cell, it seems likely that there are sufficient quantities of various negatively charged polymers to exceed the low concentrations needed in vitro to promote APC/C inhibition and activator dissociation.

The binding of activator to the APC/C is known to be regulated by multiple mechanisms. The best understood is phosphorylation of Cdh1 by cyclin-dependent kinases (Cdks), which is known to reduce Cdh1 binding to APC/C in the cell[10,12,54]. Several Cdh1 phosphorylation sites are found in N-terminal regions that mediate APC/C binding. These regions become less accessible to Cdks upon APC/C binding, suggesting that phosphorylation is likely to block association of free Cdh1 rather than promoting dissociation. Activator binding to the APC/C is also controlled in part by activator ubiquitylation. Cdc20 and Cdh1 levels decrease as a result of APC/C-mediated autoubiquitylation and degradation in late mitosis and G1, respectively[55]. Cdc20 autoubiquitylation is also involved in disruption of the mitotic checkpoint complex that blocks APC/C$^{Cdc20}$ activity during activation of the spindle assembly checkpoint[41–43,56,57]. However, there is no indication that activator phosphorylation or ubiquitylation is required for the polyanion-mediated activator dissociation we observed in cell lysates or with purified components.

In addition to providing insights into the control of APC/C-activator interactions, we uncovered clues about APC/C-substrate binding. APC/C substrates contain combinations of one to three degrons, of which the D box is particularly critical because of its role in linking the activator to the APC/C core. The amino acid sequences of the D box and other degrons vary in different substrates, and it is likely that these sequences evolved to be accommodated by different APC/C-activator complexes. Substrate affinity for each of the two activators determines the substrate residence time, which helps determine the processivity of ubiquitylation and the timing of substrate degradation in vivo[31]. Our evidence suggests that substrate affinity also controls the ability of polyanions to inhibit APC/C activity: the binding of high-affinity substrates results in resistance to the effects of polyanions, while low-affinity substrates are less resistant. Thus, differences in enzymatic activity with substrates of different affinities are amplified in the presence of

polyanions, providing a selectivity filter that promotes ubiquitylation of high-affinity targets while suppressing background activity toward low-affinity substrates. Furthermore, the ability of substrate binding to stabilize activator binding could enhance APC/C-activator function at cell-cycle stages when abundant substrates are present.

In the absence of substrate, the activator is attached with high affinity to the APC/C via flexible N- and C-terminal tethers, such that the globular WD40 domain does not interact with the Apc10 subunit and remains mobile[15,21]. D box binding bridges the gap between activator and Apc10, providing a third site of activator binding. How do polyanions promote activator dissociation? Any proposed mechanism must account for the observation that D box binding blocks the effects of DNA on both the ubiquitylation reaction and activator binding. One speculative model (Fig. 6e) is that some part of the long polyanionic polymer interacts with the APC/C at sites adjacent to the activator that are exposed in the absence of a substrate. Polyanions might push the activator WD40 domain away from Apc10, preventing D box binding (and inhibiting ubiquitylation), while at the same time promoting activator dissociation by disrupting binding sites for the activator termini. In this scenario, high-affinity D box binding would overcome the inhibitory effects of polyanions by locking the activator against Apc10.

According to this model, polyanions are essentially competitive inhibitors of D box binding. Confirmation of this idea will require the development of quantitative methods for the measurement of substrate-binding affinities at varying nucleic acid or polyphosphate concentrations. The model also predicts that D box binding should block the inhibitory interactions between polyanions and the APC/C. Interestingly, we found that increasing D box concentrations did not affect the half-maximal inhibitory DNA concentration (Fig. 6c, d), suggesting that substrate does not have much impact on polyanion binding. However, it is likely that long polyanionic molecules interact with multiple distant sites on the APC/C, and any local effects of D box binding might not result in a significant loss of overall polyanion affinity (Fig. 6e).

With combined concentrations exceeding tens of millimolar, nucleic acids, polyphosphate chains, ATP and other charged macromolecules are in frequent close contact with proteins inside the cell. Transient interactions between polyanions and protein surfaces may create further crowding in the molecular microenvironment around proteins, while also altering the structural and functional dynamics of protein complexes and other macromolecules. Emerging roles of ATP beyond energy provision suggest that the electrostatic microenvironment around biomolecules has an impact on biochemical outputs[58]. Our understanding of polyanion biology is

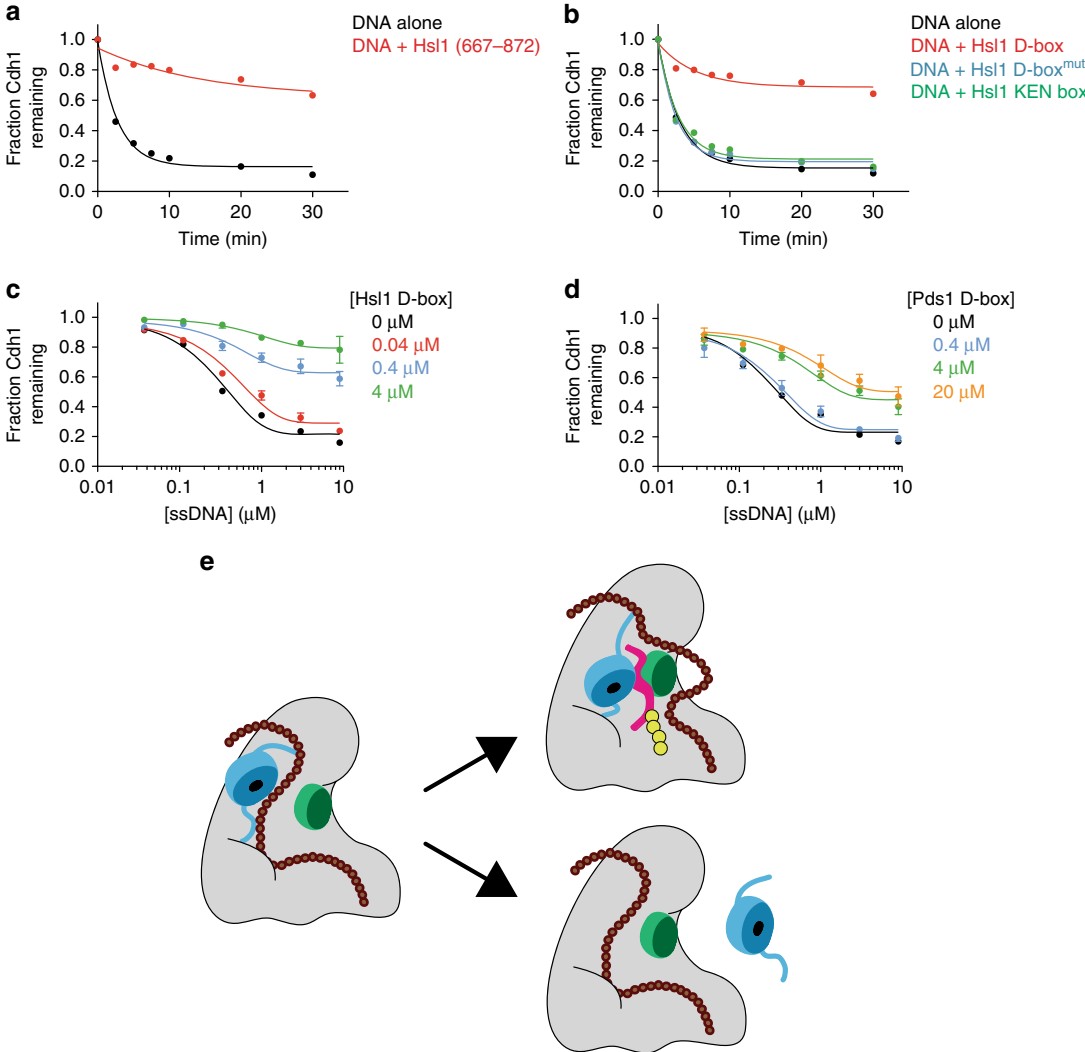

**Fig. 6 D box-dependent substrate binding protects activator from dissociation. a** Cdh1 dissociation was measured with 1.5 μM 75-base ssDNA and 3 mM ATP, in the absence (black) or presence (red) of 4 μM purified Hsl1 fragment (aa 667–872). Uncropped autoradiographs and source data are provided in the Source Data file. **b** Cdh1 dissociation was measured in the presence of buffer (black), 4 μM Hsl1 D box peptide (red), Hsl1 D box peptide mutant (blue), or Hsl1 KEN box peptide (green). Reactions were performed in the presence of 1.5 μM 75mer ssDNA and 3 mM ATP. Uncropped autoradiographs and source data are provided in the Source Data file. **c**, **d** Cdh1 dissociation was measured in the presence of different concentrations of 75 mer ssDNA and D box peptides from **c** Hsl1 or **d** Pds1 (black 0 μM, red 0.04 μM, blue 0.4 μM, green 4 μM, orange 20 μM). Reactions were supplemented with 3 mM ATP. Data indicate means (±SEM) from two independent experiments. Uncropped autoradiographs and source data are provided in the Source Data file. **e** Model for dynamic regulation of APC/C activity by polyanions. Activator (blue) binding to the APC/C is mediated by binding motifs located on flexible N- and C-termini. Long polyanionic molecules (brown) interact at multiple sites on the APC/C, perhaps disrupting interactions between the activator termini and the APC/C. We speculate that one part of the polyanion chain interferes with productive substrate D box binding by pushing the activator WD40 domain away from Apc10 (green) and/or by directly interacting with Apc10. A high-affinity D box (red) overcomes this interference and links activator to Apc10, thereby promoting activator binding and substrate ubiquitylation (yellow).

limited, but further studies could help us discover and characterize other examples of non-canonical molecular interactions that modulate complex enzyme functions.

## Methods

**Synchronization and western blotting**. All yeast strains were derivatives of W303. For western blotting of activators during the cell cycle, yeast cells carrying TAP-tagged Cdc16 (Strain yAM4: Cdc16-TAP::KanMX, MATa) were grown overnight and diluted the next morning. When cells reached OD 0.3, cells were synchronized at 30 °C for 3 h with 1 μg/ml α-factor. Following release from the arrest by washing, cells were harvested at various times and lysed by bead beating in lysis buffer as described in the next section. TAP-tagged APC/C was immunoprecipitated with magnetic IgG beads. Cdc20 and Cdh1 were subjected to western blotting with polyclonal goat yC-20 (sc-6731; Santa Cruz Biotechnology, 1:1000) and yC-16 (sc-8959; Santa Cruz Biotechnology, 1:1000), respectively.

**Activator binding and dissociation**. A yeast strain carrying Cdc16-TAP and lacking Cdh1 (Strain DOM1226: cdh1::LEU2, Cdc16-TAP:HIS, MATa)[39,41] was harvested at $OD_{600} = 1$ and flash frozen in liquid nitrogen. Cell pellets were lysed by bead beating in lysis buffer (25 mM HEPES pH 7.6, 150 mM KOAc, 2.5 mM MgCl₂, 10% Glycerol, 0.5% Triton X-100, 1 mM PMSF, 1 mM DTT, and EDTA-free protease inhibitor mix) and centrifuged at 20,000×g for 15 min at 4 °C. The APC/C was immunoprecipitated with IgG-coupled magnetic beads (Invitrogen #14301) at 4 °C. Cdc20 and Cdh1 were translated in vitro[20,39,41] using the TnT Quick-Coupled reticulocyte lysate system (Promega #L1170) and ³⁵S-Methionine (Perkin Elmer #NEG709A001MC). APC/C on beads was incubated with reticulocyte lysate containing Cdc20 or Cdh1 for 30 min at 25 °C, followed by washing three times with lysis buffer to remove unbound activator. Dissociation reactions were performed at 25 °C by incubating APC/C^Cdh1 or APC/C^Cdc20 with lysis buffer or with wild-type lysates prepared by bead beating in lysis buffer as above and supplemented with 5 mM ATP. Nucleic acid-dependent dissociation assays were performed with 3 mM ATP unless otherwise noted. To terminate the reactions, APC/C was washed with the same buffer and separated by 10% SDS–PAGE. Gels were dried on Whatman paper, exposed to a

Phosphor Screen and scanned in a Typhoon 9400 Imager. Images were analyzed using ImageQuant (GE Healthcare). Fraction of Cdc20 and Cdh1 remaining on the APC/C was calculated relative to the zero timepoint or buffer control. In Fig. 5b, APC/C was immunopurified from lysates of a strain lacking Cdh1 and carrying GFP-tagged Apc1 and TAP-tagged Cdc16 (Strain NYH14: cdh1::LEU2, Cdc16-TAP:HIS, Apc1-eGFP:CaURA3, MATa). The Hsl1 fragment used in Fig. 6a was expressed in insect cells and purified using a Strep-tag-purification column. Degron peptides used in Fig. 6b–d were purchased from CPC Scientific. Peptide sequences are listed in Supplementary Table 1.

**Partial purification of dissociation activity.** Wild-type yeast cells were collected at OD$_{600}$ = 1.0, washed twice with PBS, and resuspended in water and flash frozen in liquid nitrogen. Cells were lysed using a coffee grinder in a buffer containing 25 mM HEPES pH 7.6, 25 mM KOAc, 2.5 mM MgCl$_2$, 10% glycerol, 0.5% Triton-X 100, 1 mM PMSF, and protease inhibitor mix. Lysate was cleared by centrifugation at 100,000 × $g$ for 1 h and filtered through a 0.22 μm cellulose filter. Cleared lysate was applied to a Bio-Rad CHT type II ceramic hydro-xyapatite column using a peristaltic pump. The column was washed with 15 column volumes of buffer HA-A (25 mM HEPES 7.6, 25 mM KOAc, 10% gly-cerol) containing 2.5 mM MgCl$_2$. Samples were eluted with a step phosphate gradient using seven column volumes of the same buffer supplemented with 50, 100, 250, or 500 mM PO$_4^-$. Small portions of the flow-through and elution fractions were concentrated, buffer-exchanged into Buffer HA-A with 2.5 mM MgCl$_2$ and tested for activator dissociation in the presence of 3 mM ATP. The activity eluted at 50 and 100 mM PO$_4^-$ concentrations. The eluted activity was incubated at 95 °C for 15 min and centrifuged at 100,000 × $g$ for 1 h at 4 °C to pellet precipitated molecules. Cleared boiled samples were dialyzed into Buffer HA-A with 2.5 mM MgCl$_2$ overnight using SnakeSkin Dialysis Tubing (Thermo Fisher #68035) at 4 °C. Dialyzed samples were re-applied to the hydroxyapatite column. The vast majority of the dissociation activity did not bind to the column and was collected in the flow-through fraction. This fraction was dialyzed into Buffer HA-A overnight and concentrated 10-fold for further characterization in dissociation assays. The activity in this fraction was stable for more than 2 weeks at 4 °C with no significant loss of activity.

**RNA preparation and sequencing.** RNA species in the flow-through fraction were separated by 10% TBE-urea polyacrylamide gel, and major RNA species (~80 nucleotides) were extracted from the gel and ethanol precipitated. A 3′ DNA adapter (CTATAGTGTCACCTAAATTAATACGACTCACTATAGGG) that contains 5′ phosphate and 3′ spacers was first 5′-adenylated using a 5′-adenylation kit (NEB #E2610-S) at 65 °C for 1 h, then ligated to purified RNA species using T4 RNA Ligase 2 Truncated (NEB #M0242S) at 25 °C for 1 h. The ligation reaction was separated on a 10% TBE-urea polyacrylamide gel, and ligated products were gel purified. Next, a 5′ RNA adapter (GCAATTAACCCTCACTAAAGGAGTCGT) lacking 5′ phosphate was ligated with T4 RNA Ligase 1 (NEB #M0204S). Ligation products were gel-purified, and cDNA was synthesized using RT primer (CCCTATAGTGAGTCGTAT TAATTTAGGTGACACTATAG) and Superscript IV reverse transcriptase (Thermo-Fisher #18090010). After RNase H digestion, cDNAs were amplified using T7 (TAATACGACTCACTATAGGG) and T3 (GCAATTAACCCTCACTAAAGG) primers. PCR products were cloned into TOPO vectors and transformed into DH5-α cells. After overnight growth and plasmid isolation, sequencing was performed using M13 Forward and Reverse primers. Alignment of sequencing results with the *S. cerevisiae* S288C genome revealed the presence of tRNAs (tQ(UUG), tS(AGA), tG (CCC), tD(GUC), tE(CUC), tW(CCA), tR(CUC), tL(UAG)), and similar lengths of rRNA fragments (RDN25-1).

**Nucleic acid preparation for dissociation reactions.** Sequences used in dis-sociation assays are listed in Supplementary Table 1. RNA sequences were tran-scribed in vitro[59] using T7 RNA Polymerase (NEB #M0251). Chemically synthesized DNA oligonucleotides or transcribed RNA species were separated on a TBE-urea polyacrylamide gel, purified, ethanol precipitated, and resuspended in Buffer HA-A with or without 2.5 mM MgCl$_2$. Medium chain polyphosphate was obtained from Kerafast (p100 #EUI005) and resuspended in Buffer HA-A without MgCl$_2$.

**APC/C ubiquitylation assay.** Yeast cells carrying Cdc16-TAP and lacking Cdh1 were collected at OD$_{600}$ = 1, flash frozen, and lysed using beat beating in lysis buffer. APC/C was immunopurified on IgG beads and activated by in vitro-translated Cdc20 or Cdh1. Cdc20 and Cdh1 used in Fig. 5e–g were expressed in insect cells using the baculovirus system and purified using Strep-tag-purification methods. For APC/C substrates, Pds1, Hsl1 fragment (aa 667–872), and Clb5 were C-terminally ZZ-tagged, cloned into plasmids under control of the T7 promoter, and translated in vitro using $^{35}$S-Methionine. Substrates were purified using IgG-coupled magnetic beads and cleaved with TEV protease (Thermo Fisher #12575015). E1 and E2 (Ubc4) were expressed in *E. coli* and purified[60,61]. E2 charging was performed in a reaction containing 0.2 mg/ml Uba1, 2 mg/ml Ubc4, 2 mg/ml methylated ubiquitin (Boston Biochem #U-501) and 1 mM ATP at 37 °C for 30 min. The ubiquitylation reaction was initiated by mixing activated APC/C, E2-ubiquitin conjugates, purified substrate, and buffer

or indicated concentrations of single-stranded DNA at 25 °C. Reactions were terminated by addition of 2X SDS sample loading dye and separated by SDS–PAGE.

**Reporting summary.** Further information on research design is available in the Nature Research Reporting Summary linked to this article.

## Data availability

The source data underlying Figs. 1a, c, d, e, 2a, c, d, f, 3a–d, 4a–g, 5a–g, 6a–d, Supplementary Figs. 1b–d, 2a, b, 3a–c, and 4a–d are provided as a Source Data file. All other data are available from the corresponding author.

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

## Acknowledgements

We thank Robert Cohen, David Agard, Peter Walter, Mark Von Zastrow, and members of the Morgan lab for their technical and intellectual insights; Nairi Hartooni for providing purified Cdc20, Cdh1, and Hsl1 fragment; Kelsey Hickey for discussions and comments on the manuscript. This work was supported by the National Institute of General Medical Sciences (R35-GM118053, to D.O.M.) and an HHMI International Student Research Fellowship (A.M.).

## Author contributions

A.M. conceived the project, performed the experiments, and wrote the paper, with guidance from D.O.M.

## Competing interests

The authors declare no competing interests.

## Additional information

**Peer Review Information** *Nature Communications* thanks Ursula Jakob, Jonathon Pines and the other, anonymous, reviewer for their contribution to the peer review of this work. Peer reviewer reports are available.

