## [Peer Review File · Nature Communications]

Reviewers' comments:

Reviewer #1 (Remarks to the Author):

This study aims to shed light into the mechanism by which the anaphase promoting complex E3 ligase APC/C transiently interacts with distinct activating partners in a cell-cycle dependent manner. This manuscript describes the novel discovery that in vitro, polyanions (i.e, RNA, polyphosphate, ssDNA) effectively trigger the dissociation of activating interaction partners (i.e. Cdc20, Cdh1) from APC/C, that presence of divalent metals is inhibitory and hence requires the need for ATP or ADP in the assays , and that client binding affects the stability of the APC/C -activator complex and prevents the dissociation of the activators. The study is overall well conducted, the manuscript well written and the conclusions clear. How this will work in the physiological context of the cell, however, is still completely unclear given that cell lysates from different cell cycle stages seem not to show any differences in relative effects.

Other comments:

- 1) Many of the graphs lack error bars
- 2) Many of the experiments lack information in regards to how often the experiments were repeated
- 3) It would be interesting to know whether presence of ATP is required for polyphosphate to show its stimulatory effect on the dissociation rate
- 4) It is unclear why the authors use Mg in the first place in their assay just to then add ATP to complex it
- 5) The model that the relative affinity of the clients determines their effect on the dissociation of the activators is attractive but needs to be backed up by real numbers (i.e. Kd measurements)

Reviewer #2 (Remarks to the Author):

In this study, Mizrak and Morgan show that polyanions modulate the binding of the Cdc20 and Cdh1 co-activators with the APC/C. The authors come to this conclusion after pursuing the factor/factors that would increase the rate of dissociation of in vitro translated co-activators from APC/C immunoprecipitates to that seen in cell extracts. They provide extensive and convincing data that a variety of polyanionic species confer this activity, from RNA to DNA to polyphosphate, and that substrate - or more accurately - degen-binding represses this activity. The importance of this study is that it provides an answer to the question of how co-activator binding can be sufficiently dynamic

for the APC/C to respond to changes in the cell cycle, and may be important in determining the order in which substrates are selected by the APC/C.

Overall I recommend that the paper be published after the authors have addressed the following points.

1) The number of experimental repeats should be detailed in the figure legends and the dissociation data plotted as the mean and variation.

2) The references should be amended. Unusually for the Morgan lab, I found a number of inappropriate citations, paper omissions, and a reliance on reviews instead of primary papers.

Jonathon Pines

Reviewer #3 (Remarks to the Author):

In the manuscript by Mizrak and Morgan, the authors examine the regulation of a major E3 ubiquitin ligase, Anaphase Promoting Complex/Cyclosome (APC). The APC needs an activator for substrate ubiquitylation. However, the activators are dynamically associated with APC and have to exchange during the cell cycle. Understanding this process is fundamental to the cell cycle and additional insights into these mechanisms are surely needed. Using dissociation and ubiquitylation assays, the authors describe a mechanism where activator dissociation from APC is increased by the addition of polyanions and is competitively blocked by substrate binding. Although it is unclear if this is truly a mechanism used by the cell, the study will influence the thinking in the field when assessing the spatiotemporal control of APC activity. Therefore, the study is suitable for publication in Nature Communications. The authors should address the following points prior to publication.

1) Can the authors provide a rational or speculative example of how this mechanism could be used by the cell to promote the exchange? As the authors point out, the concentrations of polyphosphate in the cell is estimated to be far above the concentrations needed to see the observed effect in vitro. Therefore, when would the regulatory mechanism be used?

2) The finding that D-box binding inhibits polyanion-mediated activator dissociation is interesting and could be enhanced. If I understand correctly, the model suggests that polyanions and the substrate D-box compete for a binding site on the APC. Therefore, performing kinetic studies of substrate ubiquitylation by the APC in the presence and absence of polyanions should suggest that polyanions serve as competitive inhibitors. Given the lab's expertise, this seems readily feasible.

3) The authors might want to reconsider their nomenclature and interchangeable use of processive substrate versus high-affinity substrate. While processivity can be enhanced by affinity, using these terms interchangeably is misleading as other factors can also influence processivity.

Minor points:

In discussing the regulation of the binding of activators to the APC/C, it should be noted that CDH1 is also controlled through ubiquitination.

At times, for example figure 4e, it is unclear the experiments were repeated due to the lack of error bars.

Response to Reviewers

Mizrak and Morgan NCOMMS-19-20624

We thank all three reviewers for their positive and insightful comments, which have been addressed as outlined below.

Reviewer #1 (*Remarks to the Author*):

This study aims to shed light into the mechanism by which the anaphase promoting complex E3 ligase APC/C transiently interacts with distinct activating partners in a cell-cycle dependent manner. This manuscript describes the novel discovery that in vitro, polyanions (i.e, RNA, polyphosphate, ssDNA) effectively trigger the dissociation of activating interaction partners (i.e. Cdc20, Cdh1) from APC/C, that presence of divalent metals is inhibitory and hence requires the need for ATP or ADP in the assays , and that client binding affects the stability of the APC/C - activator complex and prevents the dissociation of the activators. The study is overall well conducted, the manuscript well written and the conclusions clear. How this will work in the physiological context of the cell, however, is still completely unclear given that cell lysates from different cell cycle stages seem not to show any differences in relative effects.

We thank the reviewer for the positive comments. We agree that there are no clear changes in the cell lysate activity during the cell cycle, presumably because total polyanion concentrations do not change significantly during the cell cycle. However, this does not rule out regulation of the sensitivity of the APC/C-activator complex to the activity. In particular, the ability of substrate binding to inhibit the activity raises the possibility that activator dissociation is reduced during cell cycle stages when substrates are abundant. It also remains possible that other mechanisms (APC phosphorylation?) might influence the ability of polyanions to drive activator dissociation.

We can add another important point here. Regulatory protein-protein interactions must be short-lived to provide rapid responsiveness to upstream signals. The intrinsic very high affinity of the APC/C-activator interaction is a major problem for the cell cycle control system because rapid activator dissociation is needed to provide the appropriate response to regulated changes in activator concentration or phosphorylation. Polyanions provide a constitutive mechanism to promote rapid turnover of activator, so that total activator binding declines rapidly at the appropriate times (Cdc20 in late mitosis; Cdh1 in late G1). In this respect, polyanions perform an important cell cycle function regardless of whether their activity changes or not.

Other comments:

1) Many of the graphs lack error bars

We apologize for this oversight. Many of the previous results were representative of multiple experiments, but we failed to indicate this. We have now repeated several of the key experiments and then combined results from multiple experiments to provide graphs with means and error bars. We have also constructed a very large Source Data file that provides all uncropped autoradiographs and quantifications for all experimental repeats.

2) Many of the experiments lack information in regards to how often the experiments were repeated

See point 1.

3) It would be interesting to know whether presence of ATP is required for polyphosphate to show its stimulatory effect on the dissociation rate

This is an excellent question. Our previous analysis of polyphosphate effects (Fig. 3e) was done in the presence of ATP but in the absence of magnesium, and we did not test the effect of leaving ATP out of those reactions. We have now performed new experiments to explore this question, and the new results are shown in Fig. 4e, f. As in the case of nucleic acids, the ability of polyphosphate to dissociate activator is inhibited by magnesium ions, and this inhibition is reversed by ATP. ATP is not as effective in reversing magnesium effects as it is with nucleic acids, probably because polyphosphate has a higher affinity for magnesium, such that ATP cannot compete as effectively for magnesium ions. Most importantly, ATP has no effect on polyphosphate activity in the absence of magnesium (Fig. 4f).

4) It is unclear why the authors use Mg in the first place in their assay just to then add ATP to complex it

Another excellent question, and the answer comes simply from the way the project progressed. Early in the project, when we first tested the importance of ATP for the activity (Fig. 1f), we included magnesium in the control buffer lacking ATP, to confirm that any effects were due to ATP alone. Based on this experiment, we continued to include magnesium in all buffers. It was not until very late in the project, after most experiments were completed, that we discovered that ATP was simply sequestering magnesium. Thus, most of the experiments in the paper were performed in buffers that include magnesium.

5) The model that the relative affinity of the clients determines their effect on the dissociation of the activators is attractive but needs to be backed up by real numbers (i.e. Kd measurements)

We agree that rigorous measurement of substrate binding affinities would be very helpful. However, precise quantitation of K_d values has not been possible to date because it is very difficult to prepare APC/C-activator complexes in the large quantities required for conventional binding assays. We have begun to address this problem by developing single-molecule approaches to measure substrate and degon dissociation rates, but these results are not quite ready for publication. It is therefore not yet possible to provide direct affinity measurements for the current paper. However, we did do a new series of experiments that clearly supports the idea that Hsl1 binds APC/C-Cdh1 with higher affinity than Pds1. We measured activator dissociation at varying concentrations of D-box peptides from the two substrates (new Fig. 6c, d; new Supplementary Fig. 4b, c). We find that the concentration of Hsl1 peptide required for inhibition is 10- to 50-fold lower than the required concentration of Pds1 peptide, clearly indicating that the Hsl1 peptide binds with higher affinity.

Reviewer #2 (Remarks to the Author):

In this study, Mizrak and Morgan show that polyanions modulate the binding of the Cdc20 and Cdh1 co-activators with the APC/C. The authors come to this conclusion after pursuing the factor/factors that would increase the rate of dissociation of in vitro translated co-activators from APC/C immunoprecipitates to that seen in cell extracts. They provide extensive and convincing data that a variety of polyanionic species confer this activity, from RNA to DNA to polyphosphate, and that substrate - or more accurately - degon-binding represses this activity. The importance of this study is that it provides an answer to the question of how co-activator

binding can be sufficiently dynamic for the APC/C to respond to changes in the cell cycle, and may be important in determining the order in which substrates are selected by the APC/C.

Overall I recommend that the paper be published after the authors have addressed the following points.

We thank the reviewer for his kind comments.

1) The number of experimental repeats should be detailed in the figure legends and the dissociation data plotted as the mean and variation.

As stated in our response to Reviewer #1, we have now generated plots of mean values +/- SEM for the critical experiments.

2) The references should be amended. Unusually for the Morgan lab, I found a number of inappropriate citations, paper omissions, and a reliance on reviews instead of primary papers.

We apologize for this oversight, and we have corrected and expanded the references (from 43 to 61).

Reviewer #3 (Remarks to the Author):

In the manuscript by Mizrak and Morgan, the authors examine the regulation of a major E3 ubiquitin ligase, Anaphase Promoting Complex/Cyclosome (APC). The APC needs an activator for substrate ubiquitylation. However, the activators are dynamically associated with APC and have to exchange during the cell cycle. Understanding this process is fundamental to the cell cycle and additional insights into these mechanisms are surely needed. Using dissociation and ubiquitylation assays, the authors describe a mechanism where activator dissociation from APC is increased by the addition of polyanions and is competitively blocked by substrate binding. Although it is unclear if this is truly a mechanism used by the cell, the study will influence the thinking in the field when assessing the spatiotemporal control of APC activity. Therefore, the study is suitable for publication in Nature Communications. The authors should address the following points prior to publication.

1) Can the authors provide a rational or speculative example of how this mechanism could be used by the cell to promote the exchange? As the authors point out, the concentrations of polyphosphate in the cell is estimated to be far above the concentrations needed to see the observed effect in vitro. Therefore, when would the regulatory mechanism be used?

As mentioned in our response to Reviewer #1 above, and also briefly in our Discussion, the most appealing possibility is that changes in substrate abundance govern the ability of this mechanism to drive activator dissociation. When substrates are abundant in mitosis, we speculate that their binding to the APC/C-activator complex blocks dissociation – thereby stabilizing the active enzyme precisely when it is needed. When substrate abundance declines due to degradation, polyanion-dependent dissociation becomes more effective. It is also worth mentioning again that this mechanism might not be a regulatory mechanism in the conventional sense, but rather a mechanism that simply provides the rapid activator dissociation rate that is required for rapid changes in APC/C-activator binding at specific cell cycle stages.

2) The finding that D-box binding inhibits polyanion-mediated activator dissociation is interesting and could be enhanced. If I understand correctly, the model suggests that polyanions and the substrate D-box compete for a binding site on the APC. Therefore, performing kinetic studies of substrate ubiquitylation by the APC in the presence and absence of polyanions should suggest that polyanions serve as competitive inhibitors. Given the lab's expertise, this seems readily feasible.

Yes, our preferred explanation for our results is that polyanions and D-box are competitive inhibitors of each other – although not necessarily by directly binding to the same site. We agree that the model would be supported by Michaelis-Menten analyses of apparent substrate K_M in the presence and absence of DNA. Unfortunately, our enzyme assays depend on ^{35}S -labeled substrate produced by translation in vitro, and thus the substrate concentrations we can achieve are extremely low, far below the likely K_M . It would require additional methods with alternate substrate sources to develop proper methods for measuring Pds1 K_M in the presence and absence of DNA. A more direct alternative, which we are working on but will not be ready for some time, is to determine the effects of DNA on substrate binding affinities in our new binding assays. Nevertheless, we did find a way to test the substrate-DNA competition in the other direction: that is, the ability of D-box peptides to compete with DNA. We found that increasing D-box concentrations do not seem to have an impact on the apparent affinity for DNA (Fig. 6c, d). As explained in our Discussion, we do not think this rules out the competition model. Instead, we suspect that DNA affinity for the APC/C depends on widespread multivalent interactions, and D-box binding displaces some portion of the DNA from the substrate binding site without affecting overall DNA binding affinity. These issues are certainly worth further study and we plan to explore them, but we don't feel that the current paper's central conclusions depend on complete resolution of the competition model.

3) The authors might want to reconsider their nomenclature and interchangeable use of processive substrate versus high-affinity substrate. While processivity can be enhanced by affinity, using these terms interchangeably is misleading as other factors can also influence processivity.

Good point – we have avoided doing this.

Minor points:

In discussing the regulation of the binding of activators to the APC/C, it should be noted that CDH1 is also controlled through ubiquitination.

Paragraph 4 in the Discussion mentions that both activators are regulated in part by autoubiquitination.

At times, for example figure 4e, it is unclear the experiments were repeated due to the lack of error bars.

Corrected as described above.

REVIEWERS' COMMENTS:

Reviewer #1 (Remarks to the Author):

The authors adequately addressed my previous concerns

Reviewer #3 (Remarks to the Author):

The authors have adequately addressed the points raised in the previous round of review.